# Impact on staff of providing non-invasive advanced respiratory support during the COVID-19 pandemic: a qualitative study in an acute hospital

David Wenzel  ,[1] Lucy Bleazard,[2] Eleanor Wilson,[3] Christina Faull[4]

¹Health Sciences, University of Leicester, Leicester, UK
²Leicester Medical School, University of Leicester, Leicester, UK
³Faculty of Medicine and Health Sciences, University of Nottingham, Nottingham, UK
⁴Palliative Care, LOROS Hospice, Leicester, UK

**Correspondence to**
Dr David Wenzel;
david.wenzel@nhs.net

## ABSTRACT

**Objectives** To explore the experiences of healthcare workers providing non-invasive advanced respiratory support (NARS) to critically unwell patients with COVID-19.

**Design** A qualitative study drawing on a social constructionist perspective using thematic analysis of semistructured interviews.

**Setting** A single acute UK National Health Service Trust across two hospital sites.

**Participants** Multidisciplinary team members in acute, respiratory and palliative medicine.

**Results** 21 nurses, doctors (juniors and consultants) and physiotherapists described the provision of NARS to critically unwell COVID-19 patients as extremely challenging. The main themes were of feeling ill prepared and unsupported, a need to balance complex moral actions and a sense of duty to patients and their families. The impact on staff was profound and findings are discussed via a lens of moral injury. Injurious events included staff feeling they had acted in a way that caused harm, failed to prevent harm or had been let down by seniors or the Trust. Participants identified factors that mitigated adverse impact.

**Conclusions** Although many of the issues described by participants are likely immutable components of healthcare in a pandemic, there were several important protective factors that emerged from the data. Experience, debriefing and breaks from COVID-19 wards were valuable to participants and successfully achieving a peaceful death for the patient was often viewed as compensation for a difficult journey. These protective factors may provide modelling for future education and support services to help prevent moral injury or aide in its recovery.

**Trial registration number** Registered on the Open Science Framework, DOI 10.17605/OSF.IO/TB5QJ

## INTRODUCTION

Non-invasive advanced respiratory support (NARS), including continuous positive airway pressure (CPAP) or bilevel positive airway pressure (sometimes referred to as simply non-invasive ventilation) is an essential component of modern clinical practice.[1] It is used for the treatment of critically ill patients with a range of common conditions including chronic obstructive pulmonary disease (COPD), pulmonary oedema and, more recently, COVID-19.[2][3]

NARS is used extensively in the treatment of respiratory failure due to COVID-19. It is a first-line treatment for those (largely younger people) who would be suitable for escalation to intubation should this be required and for those (largely older people) for whom it constitutes the ceiling of treatment.[3] The mortality rate is around 75% among those not considered appropriate for invasive ventilation[4] and the treatment carries significant burden.[5] A considerable number of these patients die either still using NARS or after its withdrawal, a procedure which represents a time of great uncertainty for both staff and patients.[6]

Despite this, there are little data or guidance for clinicians caring for these patients at the end of life—perhaps because of the lack of research into symptom control in this area of care.[7][8]

Quantitative studies of the impact of COVID-19 on healthcare workers have shown high levels of post-traumatic stress symptoms,[9] notably greater than in other infectious disease outbreaks.[10] This study looks to explore the causes of this impact in the highest mortality area of COVID-19 care—critically ill patients requiring NARS who were not suitable for escalation to invasive intubation.[4] There is a paucity of data both on how best to provide care for these patients

as they die and what impact this has on those who care for them.

## STRENGTHS AND LIMITATIONS OF THIS STUDY

⇒ This was a single-centre study.
⇒ DW had pre-existing relationships with study participants.
⇒ DW, under strict supervision, practiced a reflexivity-based approach for data collection and analysis with objectivity benchmarking from the interviews by LB.

## METHODS

### Design

A qualitatively designed study drawing on a social constructionist perspective using semistructured interviews.[11][12] A social constructivist perspective was used to reflect the role that participants had in forming their own truth. While each individual's truth and the interpretation presented is valid, it may lack a universality when applied to other social contexts.[13] Initially participants described a memorable patient who had used NARS and died and an iteratively developed topic guide facilitated further exploration of issues, practices and experiences.[12] This article was evaluated before publication against the Standards for Reporting Qualitative Research reporting guidelines.[14]

### Settings and participants

Doctors, nurses (from ward nursing and specialist palliative care nursing) and physiotherapists from acute medicine, respiratory medicine and palliative care were recruited using a purposive sampling approach from wards involved in the delivery of NARS in two hospital sites of an acute National Health Service (NHS) Trust in England. See online supplemental appendix A for full details of participants. In order to enable the broadest diversity of participation of staff groups participants were invited to take part individually or in pairs or groups if preferred. A maximum of 37 participants were allowed for in the study design with premature closure to recruitment to occur on thematic saturation, that is, that no further new codes were added to subsequent transcripts at the analysis stage. This saturation end point prevents unnecessary data collection that does not add to the richness of the themes.[15]

Participants had a range of experience with NARS. Four participants had less than 5 years' experience of working with NARS and 17 had more than 5 years' experience. Two of the consultant participants had worked in the founding group for introduction of NARS services to the acute hospital. Participants from the acute medicine hospital site worked within a medical unit dedicated to NARS care for the pandemic. Participants from the respiratory hospital worked either on the admissions unit or dedicated COVID-19 NARS wards.

### Patient and public involvement

No patient or public involvement was sought in the creation of this study as it focused solely on the impact of healthcare workers and contained no patient or public participation in data collection. Results from this study will be disseminated to participants via internal email.

### Data collection

In total, 19 interviews of 21 participants undertaken by telephone (n=2), Microsoft Teams (n=16) or face to face (n=1) were audio recorded (encrypted voice recorder, Olympus DS-3500) and transcribed verbatim. One group interview with three participants was performed. A summative version of the topic guide is included in online supplemental appendix B.

### Data analysis

Thematic development was contemporaneous but dynamic and in parallel with data collection based on a principles of constant comparison,[16] supported by qualitative coding software, MAXQDA Plus 2020 (V.20.4.0). Inductive coding and development of the initial coding frame was undertaken by DW and LB with iterative group (including CF and EW) discussion. The whole team sought to identify areas of agreement and commonality between participants while recognising the range of perspectives and experiences described. These, sometimes even contradictory views, were crystalised into our final data analysis and added a profound richness to the data that was facilitated by the semistructured interview format.[17] Data collection was completed when thematic saturation was achieved.

## RESULTS

In all interviews, participants discussed both the high level and the harmful consequences of the emotional, physical and psychological impact levied on them and their colleagues by the COVID-19 pandemic.

> … you know when you wear the FFP3 (FFP3 – Filtering Face Pieces 3, a tight-fitting filtration mask used as part of personal protective equipment during exposure to aerosol generating procedures.) masks? I can smell that smell [and] it takes me back to the first lady that I'd been involved in a CPAP withdrawal for and I just remember it… (Specialist palliative care nurse)

> … it massively impacted me emotionally… I got really anxious. I wasn't really sleeping. I wasn't eating much. I had a bit of talking therapy. (Junior doctor)

> I was emotionally broken. I remember crying at the end of that weekend. I was, I was like 'Oh my God' and I had to, I had to actually reach out to my consultant colleagues to say, 'I feel broken' and they were like, 'that's normal'. (Acute medicine consultant)

Five themes emerged from the data that describe the causes of such impact and factors that may protect against harmful consequences. The group interview did not yield different codes or themes to the individual interviews and therefore will not be further identified in the results.

Present throughout the data was the concept of repetition, the moral harms and impact of this work affected staff multiple times a day. This did not explain how the work was injurious and so was not incorporated into a theme, but rather underlined the impact of all the themes.

I remember when cases started coming back again recently … and immediately everyone felt that pit in their stomach… 'here we go again' and what does that mean for us? It ultimately means one thing, they're all going to die. For us, it's not necessarily a death it's the amount of death, the amount of withdrawals, the amount of things that we should see in maybe thirty years of nursing - we've had in a very compressed space of time. So, it makes you feel ill. (Palliative care nurse)

### Ill prepared and unsupported

Participants expressed concern over educational needs that were not met prior to deployment. While it was noted that a lack of education around end-of-life care and NARS, including ethics and clinical reasoning, predated the pandemic, the effects of this were compounded by COVID-19. Participants relied on informal teaching from colleagues and other sources (such as popular literature) to make up for this shortfall.

Actually, I learn most of what I know about death and dying from working with my palliative care colleagues. (Physiotherapist)

There was a perceived lack of opportunity for debriefing with most reflection after traumatic events being informal between colleagues. There was often a feeling that more could be done at an organisational level to support staff in areas with high levels of NARS COVID-19 care.

I'm not sure we got a lot from the Trust… I think there was very little formal support. I think that compared to some areas we did very badly on getting debriefs and things organized. (Acute medicine consultant)

Being short of staff and poor continuity of care were referenced by many participants as having negative impacts on their ability to provide care.

### Moral actions

Participants spoke often of actions undertaken during the pandemic that had a moral underpinning: acts that balance burden and benefit for the patient, acts that they were compelled or coerced to perform and acts they were unable to complete.

NARS was a treatment regarded as unlikely to be successful with significant treatment burden and was often felt to be started due to a lack of other options, rather than as an act in keeping with the patient's best interests. Some participants reflected that starting NARS was a way for staff to cope personally with so many deaths

You've forced a treatment that is unpleasant on somebody… And sometimes it feels like you're assaulting them… [because] people just desperately don't want another person to die. But sometimes there's just the feeling of helplessness - I think that they're feeling that [starting CPAP] is a way that they can manage

that feeling of the tsunami of dying. (Palliative care consultant)

The decision to continue NARS weighed on participants in a similar fashion and even decisions to change interface made interviewees feel complicit in this burdensome treatment.

The mask wasn't giving her much symptom relief, it wasn't giving her much support with oxygenation. So we went for the hood … to see if it made a difference A) to her symptoms and B) to her clinical condition … It didn't make a difference. (Physiotherapist)

Over half of participants reported concerns they were giving patients and their families false hope, because the chance of recovering was minimal.

Very often it's just the fact that the family are holding onto that chink of hope that something is dramatically going to turn around and the patient will improve. And you know it's not going to happen 'cause you've been there many times before. (Respiratory consultant)

Communicating the poor prognosis and the imminence of death weighed on participants' minds:

… you're looking someone in their eyes and telling them that they're dying. So that's the difficulty. (Respiratory consultant)

If CPAP treatment became too burdensome, with no chance of recovery, it was often withdrawn on the advice of senior clinicians or at the request of the patient themselves. The process of withdrawal was often described as traumatising to staff. Participants felt a responsibility to withdraw the mask and enable the patient to die as peacefully and symptom free as possible but anticipating a patient's needs and to manage any distress was challenging.

Healthcare workers had a range of feelings after withdrawals. Some reported an immense sadness. The removal of the mask emphasised the teams' perceived failure to save that person's life. Some saw the withdrawal as an opportunity to create peace and a natural death, regarding it as essential for holistic palliative care. A minority of participants reported feeling responsible for causing the death by removing the mask—but most participants discussed knowing that others felt this way, even if they themselves did not.

… being that person who takes it off, it can sometimes feel like you're the person ending that patient's life. (Physiotherapist)

The visiting restrictions participants described varied throughout the pandemic. At times only patients dying and not on NARS were allowed end-of-life visitation. The recognition of death was often felt to be delayed and late in the dying process which further complicated visiting. Participants spoke extensively about restricting visitation

from relatives during the patients' admission and especially when they were dying. This had a profound impact on some.

> But just to keep a family member away from their dying relative is probably one of the hardest things I've ever had to do. (Acute medicine consultant)

### Loss of professional autonomy

Autonomy was often felt to have been lost at the expense of managing the patients' families' expectations and readiness for the death of their loved one. It often became clear to clinicians and their teams that death was inevitable (based on oxygen requirements, ventilator dependency, work of breathing and clinical experience) before the family were ready to accept it. For patients who deteriorated despite NARS, its withdrawal was generally seen as the logical/ethical clinical decision due to its progressive loss of symptomatic relief and increasing treatment burden (claustrophobia, impact on communication, breathlessness, anxiety). There were also professional concerns around prolonging and medicalising the dying process. These approaches were sometimes in tension with family views about the value of continuing NARS:

> I think the family were more so reluctant themselves to have the CPAP mask removed. So for them I guess it provided them a sense of comfort that at least everything was done till the very end. (Ward nurse)

A huge amount of communication was dedicated to ensuring that families understood their loved one's condition and were enabled to accept the clinical recommendations. If families could not concur or be helped to be ready prior to their loved one's death an extreme amount of distress was reported.

> … they wanted his wife to be able to see him over Facetime and they had the phone with the wife, in his face, while he was dying and gasping… they had removed the CPAP…. I can remember this distinct memory of wailing and screaming and I'm just sitting there trying to make it stop and the patient died, he died, whilst the wife was in [his] face because they wanted to see his face… (Specialist palliative care nurse)

Even when families were accepting of the clinical decisions being made, staff often felt their range of options were limited due to inheriting decisions from the previous shifts or earlier in the chain of care.

> In my own mind I would think, 'why have they been put on CPAP?'. …But there wasn't somebody I could have a disagreement with because the decision had been made 2, 3, 5 days earlier. (Acute medicine consultant)

While respecting senior decision-makers and the complexity of their role, there were many references to staff being involved in clinical decisions they would not make themselves and the impact that this had on them.

> I'm probably more distressed by the people who I felt I supported a decision, or was coerced it felt like, into helping them have CPAP when I thought they were dying in the next few hours. (Palliative care consultant)

This was often due to the uncertainty of the clinical situation and at times led to conflict within the multidisciplinary team about the appropriateness of continuing therapy and prolonging or medicalising death. Disagreements were often reported as not openly discussed and the data suggest that this resulted in little awareness from senior decision-makers.

> It's quite nice getting their opinions but I wouldn't necessarily do what they wanted if it wasn't what I wanted to do. (Respiratory consultant)

> I know from a nursing perspective a lot of the nurses felt uncomfortable… I feel like a medical decision is probably very different to how the nurses felt. (Ward nurse)

Participants frequently described their difficulty in controlling the severe symptoms that were caused by the patients' illness. Their efforts were further complicated by the impact of personal protective equipment (PPE) requirements.

> And became massively tachypnoeic and was trying to sit up at that point to get his breath, which was horrific. So, and the difficulty being…you've got to put on [PPE] to get in there. (Specialist palliative care nurse)

These symptoms often occurred in patients with little history of ill health and in the context of rapid clinical decline. This made them more poignant to the staff that witnessed them.

### Duty

Exposure to pandemic experiences was often viewed in the context of a sense of duty, to patients and their families. The moral actions participants undertook were viewed in the context of a deep sense of dedication to doing the 'right' thing and a responsibility for how families would perceive the death of their loved ones

> I think it's only human to try and avoid the difficult decisions but I think it's important that we have a duty of care where we try and address things that are difficult. (Junior doctor)

> … you're guiding this experience for someone. And how are they going to look back on it? And are they going to look back on it and go, 'Yeah, that was OK.' Or are they going to look back and say, 'that was a disaster'. (Palliative care consultant)

This sense of duty led to actions by staff significantly beyond their usual role; acting as substitute family;

partaking in end-of-life religious rituals; sitting in full PPE while a patient died; or taking part in end-of-life conversations that healthcare workers are rarely party to:

> … you effectively become, the nursing staff and the medical team, effectively become the surrogate family. (Junior doctor)

> But prior to covid you wouldn't sit and listen to somebody else tell their Mum all their things that they want to in the minutes before she dies… you wouldn't witness all of that. (Palliative care consultant)

Ultimately, it appeared this sense of duty was often the driving factor for perseverance from otherwise expended staff:

> It was very much a 'we've just got to get on with this, we've just gotta get through this as best we can, remember that all of these patients are relying on us to make them comfortable'. (Palliative care nurse)

### Protective factors

Participants discussed successes they had during the pandemic and the factors that facilitated them. These seemingly buoyed their perceptions of their teams' actions and led to greater satisfaction with care provision, providing some protection from the ongoing strains of NARS care in the pandemic.

Dying peacefully was often referenced as a compensation for the overall nature of a patient's clinical journey. Ensuring minimal symptoms, family presence and every avenue of care had been explored gave participants a sense of accomplishment. The earlier clinical decisions were made and the more clearly they were communicated, the more associated they were with better patient care:

> I had a case on [a ward] who went on and did a night on CPAP and didn't tolerate it and they stopped it overnight at two o'clock in the morning. And actually, he then went home and died at home [which was his wish]. But my point more is that they sort of clocked… They didn't persevere. They clocked early that this wasn't going well. They got out." (Palliative care consultant)

Higher levels of clinical experience were associated with increased resilience and surety around clinical decision-making.

> It's a decision that you make. I can't predict the future anymore than anybody else can. But it's a case of you have to make a decision. And that's part and parcel of having been a consultant and having the luxury of experience. (Acute medicine consultant)

Regardless of whether death was peaceful or not participants spoke highly of their colleagues' support afterwards. This support was often informal, based on local team dynamics and often individuals, but felt to be a valuable part of the recovery process:

> We often talk to [name]. She is really good at asking people how they are. People like talking to her, she's very sympathetic. And then I know the matron is very much on hand to talk to the nurses. (Respiratory consultant)

> Some of the senior nurses were just fabulous, fabulous people, and we were being massively supported by them as well, as much as they would say we supported them. There were days when I was on my knees going 'I just can't come back to this,' you know. (Specialist palliative care nurse)

Some participants discussed the benefits they had received from respite away from the high acuity environment of the wards. All participants who referenced this were doctors, most consultants with other clinical or academic commitments and had concerns for others in the team who did not have this.

> It doesn't seem to have a particular effect on me but maybe I just had… less involvement over the last six months and that's allowed me to sort of lick my wounds and feel better for it. (Acute medicine consultant)

Some participants reported the use of practices to protect themselves or others from the impact of their work. Participants reported self-identifying that they were unable to complete certain tasks (mask withdrawals and breaking bad news especially) and either delaying or delegating the task and senior decision-makers also reported identifying staff who needed tasks reallocating.

> That's not always possible, but we also try to recognize that in our team… when we've done several difficult ones in a day, or in a few days, that actually you may not be the right person to do the next one a few hours later. Because you might not have enough of yourself to give. (Palliative care consultant)

### DISCUSSION

Our data demonstrated significant personal impact on participants in keeping with quantitative studies[18] but has added to this literature by identifying how working in the pandemic had this effect on staff supporting people with respiratory failure outside of intensive care units. Participants were forced by circumstance to engage in complex moral actions balancing burden and benefit while treating patients with severe symptoms. This impact was compounded by the overwhelming volume of patients and the duty staff felt to help them and their families. This complex care environment often resulted in a perceived loss of professional autonomy, further confounding participants' moral actions.

Analysis revealed significant congruence between our data and the existing canon of knowledge around the theory of moral injury. Moral injury refers to the impact felt when a person experiences events not in keeping

with their own moral code or conscience. The concept of moral injury has become closely linked to the pandemic with statements of concern about the prevalence of moral injury from governments and professional bodies.[19 20] These events are often subdivided into acts of commission, omission and betrayal.[21] Many of the actions undertaken by participants were moral actions that balanced cost or burden and benefit: restricting visiting, initiating high burden treatment and withdrawing treatment.

Being forced to balance a patient's desire and right to see their family before they die and the safety of their family and society may result in a morally 'correct' action, but the process of enacting this and the witnessed distress it causes is injurious to the practitioner. This is consistent with an act of commission, an action a person is compelled to undertake by seniors or the environment of care.[22]

The value of NARS was a balance of burdensome treatment and of hope of possible recovery and symptom relief. An inability to stop death, to control symptoms adequately or act in the best interest of the patient weighed heavily on participants. These represent acts of omission, acts that are not undertaken by healthcare workers due to the environment in which they are working, and this led participants to feel as if their care was lacking. It should be noted that there is currently no literature to support the role of NARS as effective symptom relief in COVID-19 and only limited evidence in COPD.[8]

Some of the most challenging experiences participants described were around the loss of autonomous decision-making. Feeling coerced into forcing treatment on a patient by colleagues or feeling unable to withdraw due to family readiness impacted participants who then felt complicit in the suffering that ensued. This loss of autonomy represents both an act of omission and an act of commission.

Participants expressed concern over educational needs that had not been met prior to deployment to COVID-19 wards as well as a lack of formal debriefing opportunity after challenging cases. This represents an act of betrayal, where a trusted other creates exposure to injurious events—often through lack of preparation or support.[23] However, in contrast with existing literature,[24] no references were made by participants to PPE availability and only one participant described concerns over level of exposure and the risk of catching COVID-19. Betrayal as an origin of moral injury was less prevalent across interviews than reflections on commission and omission.

Literature around moral injury describes those affected as 'witnesses'[22] to suffering. Our data suggested that the impact from injurious events was facilitated by the participants sense of ownership over the traumas they witnessed. The duty that they felt to provide good care and support to families was key in the mechanism of their suffering. This sense of 'ownership' should be considered in future works examining moral injury in healthcare workers.

Present throughout our data was the concept of repeated injury. It was not only that morally injurious events occurred but also that they occurred much more frequently than in usual circumstances. This was a significant finding present in every interview that appeared to exacerbate the impact felt by healthcare workers during the pandemic.

Moral injury represents a wide spectrum of impact, from self-limiting to chronically debilitating and requiring professional treatment.[25] This spectrum was present in our data. Some participants were unaffected, some had persistent symptoms and a minority spoke about requiring formal treatment to recover (psychological therapy).

There was great openness from participants to discuss the impact that providing end-of-life care in a pandemic had on them. This in and of itself may be indicative of the lack of formal structures in place to help staff recover. Lack of support in recovery was described frequently but there was no clear consensus over what would be most helpful in this regard.

The impact on staff of low impact/high burden treatments is rarely considered in the literature but may need to form a key part of any support packages used in the future. The low-level efficacy of NARS was anecdotally apparent to staff but not apparent in formal evidence throughout much of the pandemic and only recent papers formally question its use in 'do not intubate' patients.[3–5]

Our findings identify several issues that may form key components of preventative strategy including: education around NARS use and withdrawal,[26] the ethical reasoning around NARS and how we prepare staff for patient death. Perceived poor staff continuity and inadequate staffing levels exacerbate issues within this context but may not be easily addressed.

A great many of the components of moral injury to staff were, and remain, seemingly unavoidable. As the patients suffered, so did their caregivers. This may be a component of pandemic healthcare.[27 28] However, how we prepare, support and aide recovery warrants further investigation. Participants described several coping mechanisms, debriefing requirements and post injury recovery techniques that could aide with workforce planning moving forward.

A peaceful, and holistically provided for, death was often viewed by participants as compensation for a tumultuous journey. The role NARS may have in a peaceful death as symptom control for breathlessness is an understudied phenomenon.[29–31] Further research on the role of NARS in end-of-life care will be important to continue to improve patient care and support our staff in facilitating good end-of-life care.

## LIMITATIONS

The principle investigator, DW, is a doctor with extensive experience working in the NARS COVID-19 wards with existing relationships with the participants. While this was a considerable advantage for recruitment and openness between participant and researcher, we recognise that this could have influenced the data gathered. CF is primarily an academic palliative care physician who

spent no time in the acute trust during the pandemic. Under CF's supervision, DW practiced a reflexivity-based approach to limit the impact of personal bias on data collection and analysis—this included keeping a reflexivity journal, regular supervision meetings and splitting data collection and analysis with LB. LB had no pre-existing relationships with the research participants and limited COVID-19 exposure. LB carried out five of the interviews and analysed three independently. LB and DW jointly coded five of the transcripts and thematic generation took place in team meetings. Data were collected from only one hospital trust.

A degree of recall bias may have influenced data collection, especially as participants were asked to recall a memorable patient. However, this methodology allows significant in-depth exploration of scenarios that are impactful to the participant, which was the aim of the research.

## CONCLUSION

This study identified the considerable impacts on healthcare staff caring for patients who required NARS and died due to COVID-19. It has identified the multifactorial origins of distress and harm to staff and explores this within the context of the duty that staff felt towards their patients which augments the current understanding of moral injury.

A great many of the factors that impacted on staff are immutable and unavoidable components of severe respiratory disease. Being unable to reverse an illness process, witnessing distress and breaking bad news are regrettable essentials of this work. However, the role of staff as active perpetrators of distress – restricting visitation, starting high burden treatments, and prolonging suffering or causing death may require further reflection should the pandemic worsen, or a new one occurs. Importantly, the study was able to identify several protective factors that may prevent, reduce and aide recovery from injurious events. Understanding how to protect and minimise psychological injury to our workforce in these impactful circumstances will be essential in safeguarding the future of healthcare provision.

**Contributors** DW and CF conceptualised and developed the project. The protocol was authored by DW, who is the guarantor, and reviewed/edited by CF. Data collection tools were designed by all authors. Data collection was performed principally by DW supported by LB. Database creation was principally done by DW, supported by LB. Data analysis was performed principally by DW with support from LB, EW and CF (contribution level in order). Thematic analysis was performed equally by DW and LB with further contribution from CF and EW. The primary report was authored by DW with editing and review by EW, CF and LB (contribution level in order).

**Funding** DW's current posting, academic clinical fellow, is a funded role by the National Institute of Health Research (NIHR) (funding award reference ACF-2019-11-007). The NIHR had no input in study creation, design or reporting.

**Competing interests** None declared.

**Patient and public involvement** Patients and/or the public were not involved in the design, or conduct, or reporting, or dissemination plans of this research.

**Patient consent for publication** Not applicable.

**Ethics approval** This study involves human participants. This study was reviewed by the University Hospitals of Leicester Sponsorship Panel who determined it would not need ethics approval as it only included NHS staff members. Sponsor oversight was then provided by University Hospitals of Leicester (Edge ID: 138908). Participants gave informed consent to participate in the study before taking part.

**Provenance and peer review** Not commissioned; externally peer reviewed.

**Data availability statement** Data are available upon reasonable request. All raw materials including deidentified raw data, coding and thematic analysis can be obtained by contacting DW via email.

**ORCID iD**
David Wenzel http://orcid.org/0000-0002-3437-0785

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
